# HNPE: Leveraging Global Parameters for Neural Posterior Estimation

**Pedro L. C. Rodrigues**
Inria, CEA, Université Paris-Saclay, France
`pedro.rodrigues@inria.fr`

**Thomas Moreau**
Inria, CEA, Université Paris-Saclay, France
`thomas.moreau@inria.fr`

**Gilles Louppe**
University of Liège, Belgium
`g.louppe@uliege.be`

**Alexandre Gramfort**
Inria, CEA, Université Paris-Saclay, France
`alexandre.gramfort@inria.fr`

## Abstract

Inferring the parameters of a stochastic model based on experimental observations is central to the scientific method. A particularly challenging setting is when the model is strongly indeterminate, i.e. when distinct sets of parameters yield identical observations. This arises in many practical situations, such as when inferring the distance and power of a radio source (is the source close and weak or far and strong?) or when estimating the amplifier gain and underlying brain activity of an electrophysiological experiment. In this work, we present hierarchical neural posterior estimation (HNPE), a novel method for cracking such indeterminacy by exploiting additional information conveyed by an auxiliary set of observations sharing global parameters. Our method extends recent developments in simulation-based inference (SBI) based on normalizing flows to Bayesian hierarchical models. We validate quantitatively our proposal on a motivating example amenable to analytical solutions and then apply it to invert a well known non-linear model from computational neuroscience.

## 1 Introduction

Simulation-based inference (SBI) has the potential to revolutionize experimental science as it opens the door to the inversion of arbitrary complex non-linear computer models, such as those found in physics, biology, or neuroscience (Cranmer et al., 2020). The only requirement is to have access to a mathematical model implemented as a simulator. When applied to biophysical models and simulators in neuroscience (e.g. Leon et al. 2013), it could estimate properties of the brain closer to the cellular level, thus closing the gap between the neuroimaging and computational neuroscience communities. Grounded in Bayesian statistics, recent SBI techniques profit from recent advances in deep generative modeling to approximate the posterior distributions over the full simulator parameters. Their intrinsic quantification of uncertainties reveals whether certain parameters are worth (or not) scientific interpretation given some experimental observation.

SBI is concerned with the estimation of a conditional distribution over parameters of interest $\boldsymbol{\theta}$. Given some observation $x_0$, the goal is to compute the posterior $p(\boldsymbol{\theta}|x_0)$. It generally happens that some of these parameters are strongly coupled, leading to very structured posteriors with low dimensional sets of equally likely parameters values. For example, this happens when the data generative process depends only on the products of some parameters: multiplying one of such parameters by a constant and another by its inverse will not affect the output. Performing Bayesian inference on such models naturally leads to a "ridge" or "banana shape" in the posterior landscape, as seen e.g. in Figure 4 of Gonçalves et al. (2020). More formally the present challenge is posed as soon as the model likelihood

35th Conference on Neural Information Processing Systems (NeurIPS 2021).

function is non-injective w.r.t. $\boldsymbol{\theta}$, and is not strictly due to the presence of noise on the output observations. In statistics and econometrics literature, such models are called partially identified models (Gustafson, 2014).

To alleviate the ill-posedness of the estimation problem, one may consider a hierarchical Bayesian model (Gelman and Hill, 2007) where certain parameters are shared among different observations. In other words, the model's parameters $\boldsymbol{\theta}_i$ for an observation $\boldsymbol{x}_i$ are partitioned into $\boldsymbol{\theta}_i = \{\boldsymbol{\alpha}_i, \boldsymbol{\beta}\}$, where $\boldsymbol{\alpha}_i$ is a set of sample specific (or local) parameters, and $\boldsymbol{\beta}$ corresponds to shared (or global) parameters. For this broad class of hierarchical models, the posterior distribution for a set $\mathcal{X} = \{x_1, \ldots, x_N\}$ of $N$ observations can be written as (Tran et al., 2017):

$$p(\boldsymbol{\alpha}_1, \ldots, \boldsymbol{\alpha}_N, \boldsymbol{\beta}|\mathcal{X}) \propto p(\boldsymbol{\beta}) \prod_{i=1}^{N} p(x_i|\boldsymbol{\alpha}_i, \boldsymbol{\beta})p(\boldsymbol{\alpha}_i|\boldsymbol{\beta}). \tag{1}$$

Hierarchical models share statistical strength across observations, hence resulting in sharper posteriors and more reliable estimates of the (global and local) parameters and their uncertainty. Examples of applications of hierarchical models are topic models (Blei et al., 2003), matrix factorization algorithms (Salakhutdinov et al., 2013), including Bayesian non-parametrics strategies (Teh and Jordan, 2010), and population genetics (Bazin et al., 2010).

In this work, we further assume that the likelihood function $p(x_i|\boldsymbol{\alpha}_i, \boldsymbol{\beta})$ is implicit and intractable, which implies that traditional MCMC methods can not be used to estimate the posterior distribution. This setup leads to so-called likelihood-free inference (LFI) problems and many algorithms (Papamakarios and Murray, 2016; Greenberg et al., 2019; Hermans et al., 2020; Durkan et al., 2020b) have recently been developed to carry out inference under this scenario. These methods all operate by learning parts of the Bayes' rule, such as the likelihood function, the likelihood-to-evidence ratio, or the posterior itself. Approaches for LFI in hierarchical models exist, but are limited. Bazin et al. (2010) extend approximate Bayesian computation (ABC) into a two-step procedure in which local and global variables are estimated. Tran et al. (2017) adapt variational inference to hierarchical implicit models using a GAN-like training approach, while Brehmer et al. (2019) and Hermans et al. (2020) extend amortized likelihood ratios to deal with global parameters, but cannot do inference on local parameters. Motivated by the posterior estimates of individual samples, we consider a sequential neural posterior estimation approach derived from SNPE-C (Greenberg et al., 2019).

The paper is organized as follows. First, we formalize our estimation problem by introducing the notion of global and local parameters, and instantiate it on a motivating example amenable to analytic posterior estimates allowing for quantitative evaluation. Then, we propose a neural posterior estimation technique based on a pair of normalizing flows and a *deepset* architecture (Zaheer et al., 2017) for conditioning on the set $\mathcal{X}$ of observations sharing the global parameters; we call our method 'hierarchical neural posterior estimation', or simply HNPE. Results on an application with time series produced by a non-linear model from computational neuroscience (Ableidinger et al., 2017) demonstrate the gain in statistical power of our approach thanks to the use of auxiliary observations. We also use this model to analyse real brain signals, giving a full demonstration of the power of LFI to relate parameters from theoretical models to real experimental recordings.

## 2 Hierarchical models with global parameters

**Motivating example.** Consider a stochastic model with two parameters, $\alpha$ and $\beta$, that generates as output $x = \alpha\beta + \varepsilon$, where $\varepsilon \sim \mathcal{N}(0, \sigma^2)$. We assume that both parameters have uniform prior distribution $\alpha, \beta \sim \mathcal{U}[0, 1]$ and that $\sigma$ is known and small. Our goal is to obtain the posterior distribution of $(\alpha, \beta)$ for a given observation $x_0 = \alpha_0\beta_0 + \varepsilon$. This simple example describes common situations where indeterminacy emerges. For instance, $x_0$ could be the radiation power measured by a sensor, $\alpha$ the intensity of the emitting source, and $\beta$ the inverse squared distance of the sensor to the source. In this case, a given measurement may have been due to either close weak sources ($\alpha \downarrow$ and $\beta \uparrow$) or far strong ones ($\alpha \uparrow$ and $\beta \downarrow$). Using Bayes' rule and considering $\sigma$ small we can write (see Appendix A for more details)

$$p(\alpha, \beta|x_0) \approx \frac{e^{-(x_0 - \alpha\beta)^2/2\sigma^2}}{\sqrt{2\pi\sigma^2}} \frac{\mathbf{1}_{[0,1]}(\alpha)\mathbf{1}_{[0,1]}(\beta)}{\log(1/x_0)}, \tag{2}$$

where $\mathbf{1}_{[a,b]}(x)$ is an indicator function that equals one for $x \in [a, b]$ and zero elsewhere. Note that the first term in the product converges to $\delta(x_0 - \alpha\beta)$ as $\sigma \to 0$ and that the joint posterior distribution has

an infinite number of pairs $(\alpha, \beta)$ with the same probability, revealing the parameter indeterminacy of this example. Indeed, for $x \in [0, 1]$ and $\beta \in [x, 1]$, all pairs of parameters $(\frac{x}{\beta}, \beta)$ yield the same observations and the likelihood function $p(\cdot|\frac{x}{\beta}, \beta)$ is constant. Thus, the posterior distribution has level sets with a ridge or "banana shape" along these solutions. The top row of Figure 1 on Page 4 portrays the joint and the marginal posterior distributions when $(\alpha_0, \beta_0) = (0.5, 0.5)$ and $\sigma = 0$.

**Exploiting the additional information in $\mathcal{X}$.** Our motivating example illustrates a situation where two parameters are related in such a way that one may not be known without the other. In practice, however, it is possible that one of these parameters is shared with other observations. For instance, this is the case when a single source of radiation is measured with multiple sensors located at different unknown distances. The power of the source is fixed across multiple measurements and its posterior can be better inferred by aggregating the information from all sensors. Our goal in this section is to formalize such setting so as to leverage this additional information and obtain a posterior distribution that 'breaks' parameter indeterminacy. Note that the root cause of the statistical challenge here is not the presence of noise, but rather the intrinsic structure of the observation model.

To tackle the inverse problem of determining the posterior distribution of parameters $(\boldsymbol{\alpha}_0, \boldsymbol{\beta})$ given an observation $x_0$ of a stochastic model, we consider the following scenario. We assume that the model's structure is such that $\boldsymbol{\alpha}_0$ is a parameter specific to each observation (local), while $\boldsymbol{\beta}$ is shared among different observations (global). Yet, both are unknown. We consider having access to a set $\mathcal{X} = \{x_1, \ldots, x_N\}$ of additional observations generated with the same $\boldsymbol{\beta}$ as $x_0$.

Taking the model's hierarchical structure into account we use Bayes' rule to write

$$
\begin{aligned}
p(\boldsymbol{\alpha}_0, \boldsymbol{\beta}|x_0, \mathcal{X}) &= p(\boldsymbol{\alpha}_0|\boldsymbol{\beta}, x_0, \mathcal{X})p(\boldsymbol{\beta}|x_0, \mathcal{X}) \\
&\propto p(\boldsymbol{\alpha}_0|\boldsymbol{\beta}, x_0)p(x_0, \mathcal{X}|\boldsymbol{\beta})p(\boldsymbol{\beta}) \\
&\propto p(\boldsymbol{\alpha}_0|\boldsymbol{\beta}, x_0)p(\boldsymbol{\beta}) \prod_{i=0}^{N} p(x_i|\boldsymbol{\beta}) \\
&\propto p(\boldsymbol{\alpha}_0, \boldsymbol{\beta}|x_0)p(\boldsymbol{\beta})^{-N} \prod_{i=1}^{N} p(\boldsymbol{\beta}|x_i)
\end{aligned}
\tag{3}
$$

which shows how the initial posterior distribution $p(\boldsymbol{\alpha}_0, \boldsymbol{\beta}|x_0)$ is modified by additional observations from $\mathcal{X}$ sharing the same $\boldsymbol{\beta}$ as $x_0$. In Section 3, we present a strategy for approximating such posterior distribution when the likelihood function of the stochastic model of interest is intractable and, therefore, the posterior distributions $p(\boldsymbol{\alpha}_0|\boldsymbol{\beta}, x_0)$ and $p(\boldsymbol{\beta}|x_0, \mathcal{X})$ have to be approximated with conditional density estimators.

**Motivating example with multiple observations.** We now detail the effect of $\mathcal{X}$ on the posterior distribution of our motivating example. The $N + 1$ observations in $\{x_0\} \cup \mathcal{X}$ are such that $x_i = \alpha_i \beta_0 + \varepsilon$ for $i = 0, \ldots, N$ with $\alpha_i \sim \mathcal{U}[0, 1]$ drawn from the same prior. The posterior distribution may be written as (see Appendix A)

$$
p(\alpha_0, \beta|x_0, \mathcal{X}) \approx p(\alpha_0, \beta|x_0)\frac{\mathbf{1}_{[\mu, 1]}(\beta)}{\beta^N}\frac{N \log(1/x_0)}{(1/\mu^N - 1)} \ ,
\tag{4}
$$

where $\mu = \max(\{x_0\} \cup \mathcal{X})$. This expression shows how the initial full posterior distribution (2) changes with the extra information conveyed by $\mathcal{X}$. It can be also shown that as $N \to 0$ (no additional observations) the posterior distribution converges back to $p(\alpha_0, \beta|x_0)$. Figure 1 portrays the joint and marginal posterior distributions with $N = 10$ and $N = 100$.

## 3 HNPE : neural posterior estimation on Bayesian hierarchical models

When the likelihood function of the stochastic model is intractable, MCMC methods commonly used for posterior estimation are not applicable, since they depend on the evaluation of likelihood ratios, which are not available analytically nor numerically. To bypass such difficulty, we employ tools from likelihood-free inference (LFI) to directly estimate an approximation to the posterior distribution using a conditional neural density estimator trained over simulations of the model. In what follows, we present a novel neural network architecture for approximating the posterior distribution of a hierarchical model with global parameters based on normalizing flows. We also describe the training procedure for learning the parameters of the network using a multi-round procedure known as sequential neural posterior estimation or SNPE-C (Greenberg et al., 2019).

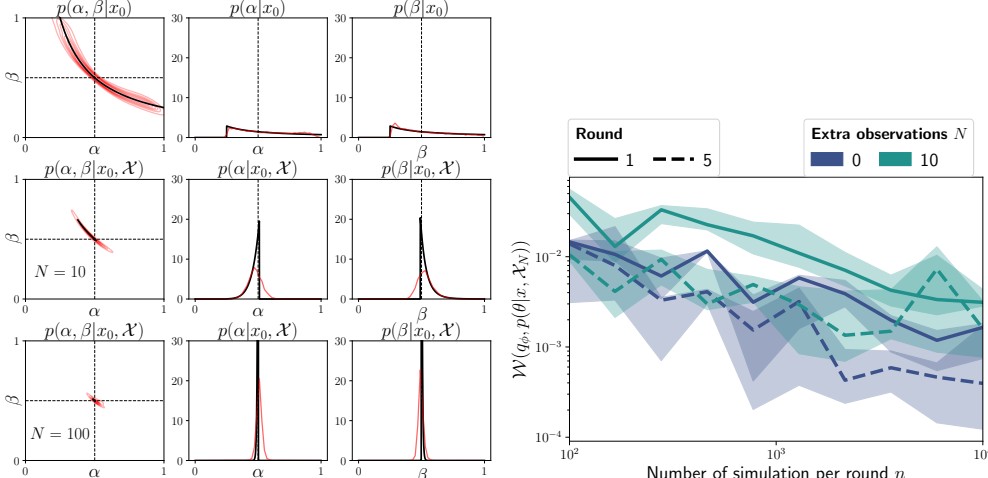

Figure 1: Results on the motivating example from Section 2 with $\sigma = 0$. (**Left**) Plots of the analytic (black) and approximated (red) posterior distributions; the ground truth values $\alpha_0$ and $\beta_0$ which generate $x_0$ are indicated with dashed lines. Approximations are obtained using the strategy described in Section 3 with $R = 5$ rounds of $n = 10^4$ simulations from the model. We observe that adding $N = 10$ and $N = 100$ observations to $\mathcal{X}$ significantly reduces uncertainty over the estimates of $\alpha_0$ and $\beta_0$. (**Right**) Evolution of the Sinkhorn divergence $\mathcal{W}_\epsilon$ between the analytic posterior distribution $p(\theta|x_0, \mathcal{X})$ and our approximation $q_\phi$ trained with an increasing number of simulations per round. The larger the simulation budget, the closer the learned posterior is to the analytic one for any number of extra observations. Sequential refinement of the posterior (*dash*) improves the approximation.

## 3.1 Approximating the posterior distribution with two normalizing flows

We approximate $p(\boldsymbol{\alpha}_0, \boldsymbol{\beta}|x_0, \mathcal{X})$ based on its factorization (3) as follows:

$$
\begin{aligned}
p(\boldsymbol{\beta}|x_0, \mathcal{X}) &\approx q_{\boldsymbol{\phi}_1}(\boldsymbol{\beta}|x_0, f_{\boldsymbol{\phi}_3}(\mathcal{X})) \\
p(\boldsymbol{\alpha}_0|\boldsymbol{\beta}, x_0) &\approx q_{\boldsymbol{\phi}_2}(\boldsymbol{\alpha}_0|\boldsymbol{\beta}, x_0)
\end{aligned}
\tag{5}
$$

where $q_{\boldsymbol{\phi}_1}$ and $q_{\boldsymbol{\phi}_2}$ are normalizing flows, i.e., invertible neural networks capable of transforming data points sampled from a simple distribution, e.g. Gaussian, to approximate any probability density function (Papamakarios et al., 2019). The function $f_{\boldsymbol{\phi}_3}$ is a *deepset* neural network (Zaheer et al., 2017) structured as $f_{\boldsymbol{\phi}_3}(\mathcal{X}) = g_{\boldsymbol{\phi}_3^{(1)}} \left( \frac{1}{N} \sum_{i=1}^{N} h_{\boldsymbol{\phi}_3^{(2)}}(x_i) \right)$, where $h$ is a neural network parametrized by $\boldsymbol{\phi}_3^{(1)}$ that generates a new representation for the data points in $\mathcal{X}$ and $g$ is a network parametrized by $\boldsymbol{\phi}_3^{(2)}$ that processes the average value of the embeddings. Note that this aggregation step is crucial for imposing the invariance to permutation of the neural network. It would also be possible to choose other permutation invariant operations, such as the maximum value of the set or the sum of its elements, but we have observed more stable performance on our experiments when aggregating the observations by their average. It is possible to show that $f_{\boldsymbol{\phi}_3}$ is an universal approximator invariant to the ordering of its inputs (Zaheer et al., 2017). Such property is important for our setting because the ordering of the extra observations in $\mathcal{X}$ should not influence the approximation of the posterior distribution. We refer to our approximation either by its factors $q_{\boldsymbol{\phi}_1}$ and $q_{\boldsymbol{\phi}_2}$ or by $q_{\boldsymbol{\phi}}$ with $\boldsymbol{\phi} = \{\boldsymbol{\phi}_1, \boldsymbol{\phi}_2, \boldsymbol{\phi}_3\}$.

**Estimating $\phi$.** We estimate $\phi$ by minimizing the average Kullback-Leibler divergence between the posterior distribution $p(\boldsymbol{\alpha}_0, \boldsymbol{\beta}|x_0, \mathcal{X})$ and $q_{\boldsymbol{\phi}}(\boldsymbol{\alpha}_0, \boldsymbol{\beta}|x_0, \mathcal{X})$ for different $x_0$ and $\mathcal{X}$:

$$
\min_{\boldsymbol{\phi}} \ \mathbb{E}_{p(x_0, \mathcal{X})} \left[ \mathrm{KL}(p(\boldsymbol{\alpha}_0, \boldsymbol{\beta}|x_0, \mathcal{X}) \| q_{\boldsymbol{\phi}}(\boldsymbol{\alpha}_0, \boldsymbol{\beta}|x_0, \mathcal{X})) \right] \ ,
$$

where $\mathrm{KL}(p \| q_{\boldsymbol{\phi}}) = 0$ if, and only if, $p(\boldsymbol{\alpha}_0, \boldsymbol{\beta}|x_0, \mathcal{X}) = q_{\boldsymbol{\phi}}(\boldsymbol{\alpha}_0, \boldsymbol{\beta}|x_0, \mathcal{X})$. We may rewrite the optimization problem in terms of each of its parameters to get

$$
\min_{\boldsymbol{\phi}_1, \boldsymbol{\phi}_2, \boldsymbol{\phi}_3} \ \mathcal{L}_{\boldsymbol{\alpha}}(\boldsymbol{\phi}_2) + \mathcal{L}_{\boldsymbol{\beta}}(\boldsymbol{\phi}_1, \boldsymbol{\phi}_3)
\tag{6}
$$

with

$$\mathcal{L}_{\boldsymbol{\alpha}}(\boldsymbol{\phi}_2) = -\mathbb{E}_{p(x_0, \mathcal{X}, \boldsymbol{\alpha}_0, \boldsymbol{\beta})} \left[ \log(q_{\boldsymbol{\phi}_2}(\boldsymbol{\alpha}_0 | \boldsymbol{\beta}, x_0)) \right] \ ,$$
$$\mathcal{L}_{\boldsymbol{\beta}}(\boldsymbol{\phi}_1, \boldsymbol{\phi}_3) = -\mathbb{E}_{p(x_0, \mathcal{X}, \boldsymbol{\alpha}_0, \boldsymbol{\beta})} \left[ \log(q_{\boldsymbol{\phi}_1}(\boldsymbol{\beta} | x_0, f_{\boldsymbol{\phi}_3}(\mathcal{X}))) \right] \ .$$

**Training from simulated data.** In practice, we minimize the objective function in (6) using a Monte-Carlo approximation with data points generated using the factorization $p(x_0, \mathcal{X}, \boldsymbol{\alpha}_0, \boldsymbol{\beta}) = p(\boldsymbol{\beta}) \prod_{i=0}^{N} p(x_i | \boldsymbol{\alpha}_i, \boldsymbol{\beta}) p(\boldsymbol{\alpha}_i | \boldsymbol{\beta})$ where $p(\boldsymbol{\alpha}_i, \boldsymbol{\beta}) = p(\boldsymbol{\alpha}_i | \boldsymbol{\beta}) p(\boldsymbol{\beta})$ is a prior distribution describing our initial knowledge about the parameters, and $p(x_i | \boldsymbol{\alpha}_i, \boldsymbol{\beta})$ is related to the stochastic output of the simulator for a given pair of parameters $(\boldsymbol{\alpha}_i, \boldsymbol{\beta})$. More concretely, the training dataset is generated as follows: First, sample a set of parameters from the prior distribution such that $(\boldsymbol{\alpha}_i^j, \boldsymbol{\beta}^j) \sim p(\boldsymbol{\alpha}_i, \boldsymbol{\beta})$ with $j = 1, \ldots, n$ and $i = 0, \ldots, N$. Then, for each $(i, j)$-pair, generate an observation from the stochastic simulator $x_i^j \sim p(x | \boldsymbol{\alpha}_i^j, \boldsymbol{\beta}^j)$ so that each observation $x_0^j$ is accompanied by its corresponding $N$ extra observations $\mathcal{X}^j = \{x_1^j, \ldots, x_N^j\}$. The losses $\mathcal{L}_{\boldsymbol{\alpha}}$ and $\mathcal{L}_{\boldsymbol{\beta}}$ are then approximated by

$$\mathcal{L}_{\boldsymbol{\alpha}}^n = -\tfrac{1}{n} \sum_{j=1}^n \log(q_{\boldsymbol{\phi}_2}(\boldsymbol{\alpha}_0^j | \boldsymbol{\beta}^j, x_0^j)) \quad \text{and} \quad \mathcal{L}_{\boldsymbol{\beta}}^n = -\tfrac{1}{n} \sum_{j=1}^n \log(q_{\boldsymbol{\phi}_1}(\boldsymbol{\beta}^j | x_0^j, f_{\boldsymbol{\phi}_3}(\mathcal{X}^j))) \ .$$

### 3.2 Refining the approximation with multiple rounds

The optimization strategy above minimizes the KL divergence between the true posterior distribution $p$ and the approximation $q_{\boldsymbol{\phi}}$, on average, for all possible values of $x_0$ and $\mathcal{X}$. This is sometimes called amortization, since the posterior distribution is expected to be well approximated for every possible observation. However, when the observed data is scarce and/or difficult to obtain or simulations of the model are costly, it might be useful to focus the capacity of $q_{\boldsymbol{\phi}}$ to better estimate the posterior distribution for a specific choice of $\tilde{x}_0$ and $\tilde{\mathcal{X}}$.

We target the approximation $q_{\boldsymbol{\phi}}$ to $\tilde{x}_0$ and $\tilde{\mathcal{X}}$ using an adaptation to SNPE-C (Greenberg et al., 2019). This algorithm uses a multiround strategy in which the data points used for minimizing the loss function $\mathcal{L}$ and obtaining parameters $\boldsymbol{\phi}^{(r)}$ at round $r$ are obtained from simulations with $\boldsymbol{\alpha}_0, \boldsymbol{\beta} \sim q_{\boldsymbol{\phi}^{(r-1)}}(\boldsymbol{\alpha}_0, \boldsymbol{\beta} | \tilde{x}_0, \tilde{\mathcal{X}})$. At round $r = 0$, parameters $\boldsymbol{\alpha}_0$ and $\boldsymbol{\beta}$ are generated from their prior distributions, which boils down to the procedure described in Section 3.1. Note that an important point is that for the different rounds, the extra observations $\mathcal{X}$ should be simulated with the parameters $\boldsymbol{\alpha}_i^j$ drawn from the original prior distribution $p(\boldsymbol{\alpha}_i | \boldsymbol{\beta})$, since the posterior distribution returned by the multi-round procedure is only targeted for observation $\tilde{x}_0$. We refer the reader to Greenberg et al. (2019) for further details on the usual SNPE-C procedure, notably a proof of convergence (which extends to our case) of the targeted version of $q_{\boldsymbol{\phi}}$ to the correct posterior density $p(\boldsymbol{\alpha}_0, \boldsymbol{\beta} | \tilde{x}_0, \tilde{\mathcal{X}})$ as the number of simulations per round tends to infinity. Algorithm 1 describes the procedure for obtaining $q(\boldsymbol{\alpha}_0, \boldsymbol{\beta} | \tilde{x}_0, \tilde{\mathcal{X}})$ after $R$ rounds of $n$ simulations.

---

**Algorithm 1:** Sequential posterior estimation for hierarchical models with global parameters

---

**Input :** observation $\tilde{x}_0, \tilde{\mathcal{X}}$, prior $p^{(0)}$, simulator $\mathcal{S}$

1 **for** *round* $r = 1$ **to** $R$ **do**
2      **for** *sample* $j = 1$ **to** $n$ **do**
3          Draw $x_0^j = \mathcal{S}(\boldsymbol{\alpha}_0^j, \boldsymbol{\beta})$ for $(\boldsymbol{\alpha}_0^j, \boldsymbol{\beta}^j) \sim p^{(r-1)}$;
4          Draw a set of extra observations $\mathcal{X}^j = \left\{ \mathcal{S}(\boldsymbol{\alpha}_i^j, \boldsymbol{\beta}^j) \text{ for } \boldsymbol{\alpha}_i^j \sim p^{(0)}(\cdot | \boldsymbol{\beta}^j) \right\}_{i=1}^N$;
5      Train $q_{\boldsymbol{\phi}^{(r)}}$ to minimize $\mathcal{L}_{\boldsymbol{\alpha}}^n + \mathcal{L}_{\boldsymbol{\beta}}^n$;
6      Set next proposal $p^{(r)} = q_{\boldsymbol{\phi}^{(r)}}(\cdot | \tilde{x}_0, \tilde{\mathcal{X}})$;
7 **return** *posterior* $q_{\boldsymbol{\phi}^{(R)}}(\cdot | \tilde{x}_0, \tilde{\mathcal{X}})$

---

## 4 Experiments

All experiments described next are implemented with Python (Python Software Fundation, 2017) and the `sbi` package (Tejero-Cantero et al., 2020) combined with PyTorch (Paszke et al., 2019), Pyro (Bingham et al., 2018) and `nflows` (Durkan et al., 2020a) for posterior estimation[1]. In all

---

[1]Code is available in the supplementary materials.

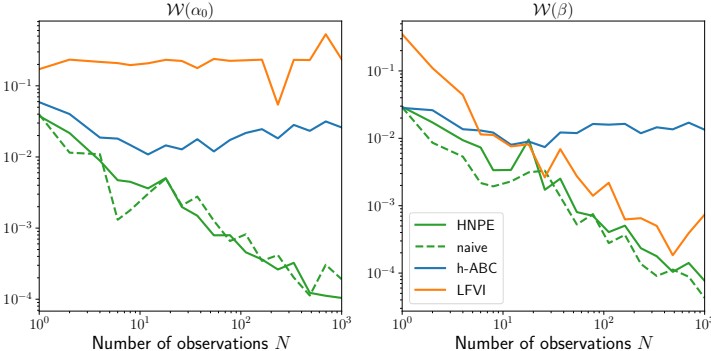

Figure 2: Results on the motivating example described in Section 2 ($\sigma = 0.05$). We see that the marginal posteriors get sharper around the ground truth parameter when more observations are available.

experiments, we use the Adam optimizer (Kingma and Ba, 2014) with default parameters, a learning rate of $5.10^{-4}$ and a batch size of 100. The code required for reproducing most of the results presented in the paper is available at `https://github.com/plcrodrigues/HNPE`

## 4.1 Results on the motivating example

To evaluate the impact of leveraging multiple observations when estimating the parameters of a hierarchical model, we use the model presented in Section 2, where the observation $x_0$ is obtained as the product of two parameters $\alpha_0$ and $\beta_0$ with independent uniform prior distributions in $[0, 1]$ (we consider the case where $\sigma = 0$). The set of extra observations $\mathcal{X} = \{x_i\}_{i=1}^N$ is obtained by fixing the same global parameter $\beta_0$ for all $x_i$ and sampling local parameters $\alpha_i$ from the prior distribution.

Our approximation to the posterior distribution consists of two conditional neural spline flows of linear order (Durkan et al., 2019), $q_{\phi_1}$ and $q_{\phi_2}$, both conditioned by dense neural networks with one layer and 20 hidden units. We use neural spline flows because of the highly non-Gaussian aspect of the analytic marginal posterior distributions, which can be well captured by this class of normalizing flows. In general, however, the true posterior distribution is not available, so using other classes of normalizing flows might be justifiable, especially if one's main goal is simply to identify a set of parameters generating a given observation. We set the function $f_{\phi_3}$ to be simply an averaging operation over the elements of $\mathcal{X}$ as the observations in this case are scalar, so the only parameters to be learned in Algorithm 1 are $\phi_1$ and $\phi_2$.

We first illustrate in Figure 1 the analytic posterior distribution $p(\alpha, \beta|x_0, \mathcal{X})$ and the approximation $q_\phi(\alpha, \beta|x_0, \mathcal{X})$ with an increasing number of extra observations ($N = 0, 10, 100$). For $N = 0$, i.e. only $x_0$ is available, we observe a ridge shape in the joint posterior distribution, which is typical of situations with indeterminacies where all solutions $(\frac{x}{\beta}, \beta)$ have the same probability. The addition of a few extra observations resolves this indeterminacy and concentrates the analytic posterior distribution on a reduced support $[x_0, \min(1, \frac{x_0}{\mu})] \times [\mu; 1]$, where $\mu = \max(\{x_0\} \cup \mathcal{X})$. Moreover, on this support, the solutions are no longer equally probable due to the $\beta^{-N}$ factor that increases the probability of solutions close to $\mu$. Also note that our estimated posterior is close to the analytic one in all cases.

To have a quantitative evaluation of the quality of our approximations $q_\phi$, in Figure 1 we plot the Sinkhorn divergence (Feydy et al., 2019) $\mathcal{W}_\epsilon$ for $\epsilon = 0.05$ between the analytical posterior $p(\alpha, \beta|x_0, \mathcal{X})$ and our approximation for different numbers of simulations per round (cf. Algorithm 1). The curves display the median value for nine different choices of $(\alpha_0, \beta_0)$ and the transparent area represent the first and the third quartiles. As expected, we note that as the number of simulations per round increases, the approximation gets closer to the analytic solution. The figure also confirms the intuition that, in general, the sequential refinement of multiple rounds leads to better approximations of the true posterior distribution for a fixed observation.

Our next analysis assesses how the posterior approximation concentrates around a given point in the $(\alpha_0, \beta)$ space as the number of extra observations $N$ increases. In Figure 2, we display the Wasserstein distances between the marginals of the learned posterior distribution and a Dirac at the ground truth values generating the observation $x_0$; we consider the results on nine different choices of $(\alpha_0, \beta_0)$ but display only the median results. We see that the distance to the Dirac for the global parameter $\beta$ decreases as more observations are added to $\mathcal{X}$, but for the local parameter $\alpha_0$ it stabilizes

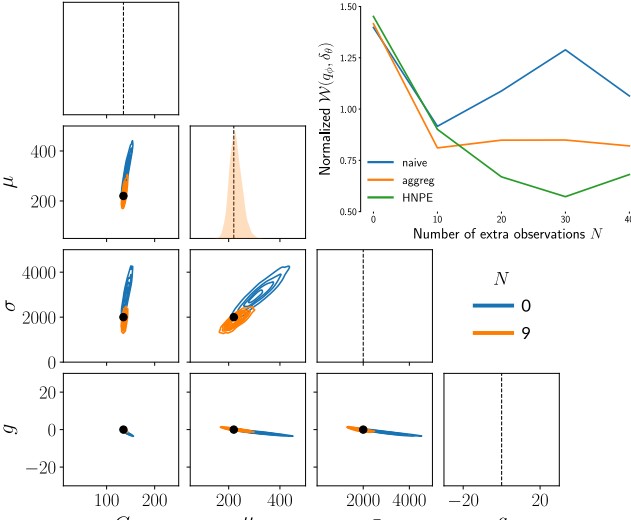

Figure 3: Posterior estimates for the parameters of the neural mass model obtained on 8 s of data sampled at 128 Hz and simulated using $C = 135$, $\mu = 220$, $\sigma = 2000$, and $g = 0$ (represented with black dots in the figure). We observe that the concentration of the posterior distribution around a Dirac converges to a lower bound when $N$ grows, reflecting an intrinsic uncertainty in the parameter estimation that cannot be reduced with more observations.

on a lower bound. This happens because $\beta$ is observed several times and, therefore, expected to be well estimated, whereas the local parameter is obtained "through the lens" of the estimated $\beta$ with information from a single observation $x_0$ corrupted by additive noise ($\alpha_0 \approx x_0/\beta$). We compare our method (HNPE) with three other approaches: a naive posterior estimation using a single normalizing flow with the same capacity as the approximation with $q_{\phi_1}$ and $q_{\phi_2}$, i.e. two layers with 20 hidden units each, in which we stack the observations from $x_0$ and the average from those in $\mathcal{X}$ as context variables, the hierarchical ABC ($h$-ABC) proposed in Bazin et al. (2010) and the likelihood-free variational inference (LFVI) presented in Tran et al. (2017). The flow-based approaches are trained with $R = 5$ rounds of $10^4$ simulations and $h$-ABC has the same simulation budget with acceptance rate of 1%. We see that the naive approach has very similar performance to HNPE, mainly due to the low dimensionality of the example being considered (in Section 4.2 we show an example where the naive architecture has similar performance to HNPE as well but taking much longer to train). LFVI captures well the global parameter as $N$ increases, but it performs poorly for the local parameter. Indeed, we have not found any evidence in the literature showing that LFVI could well estimate local parameters. For instance, all examples in Tran et al. (2017) involve only global variables. The posterior estimated with $h$-ABC does not concentrate for any of the parameters, which indicates that it would probably need a larger simulation budget to attain results comparable to the other methods.

## 4.2 Inverting a non-linear model from neuroscience

We consider a class of non-linear models from computational neuroscience known as *neural mass models* (Jansen and Rit, 1995) (NMM). These models of cortical columns consist of a set of physiologically motivated stochastic differential equations able to replicate oscillatory electrical signals observed with electroencephalography (EEG) or using intracranial electrodes (Deco et al., 2008). Such models are used in large-scale simulators (Sanz Leon et al., 2013) to generate realistic neural signals oscillating at different frequencies and serve as building blocks for several simulation studies in cognitive and clinical neuroscience (Aerts et al., 2018). In what follows, we focus in the stochastic version of such models presented in Ableidinger et al. (2017) and use the C++ implementation in the supporting code of Buckwar et al. (2019). In simple terms, the NMM that we consider may be seen as a generative model taking as input a set of four parameters and generating as output a time series $x$. The parameters of the neural mass model are:

- $C$, which represents the degree of connectivity between excitatory and inhibitory neurons in the cortical column modelled by the NMM. This connectivity is at the root of the temporal behavior of $x$ and only certain ranges of values generate oscillations.

- $\mu$ and $\sigma$ model the statistical properties of the incoming oscillations from other neighbouring cortical columns. They drive the oscillations of the NMM and their amplitudes have a direct effect on the amplitude of $x$.

- $g$ represents a gain factor relating the amplitude of the physiological signal $s$ generated by the system of differential equations for a given set $(C, \mu, \sigma)$, and the electrophysiology measurements $x$, expressed in Volts.

The reader is referred to Appendix B for the full description of the stochastic differential equations defining the neural mass model.

Note that the NMM described above suffers from indeterminacy: the same observed signal $x_0$ could be generated with larger (smaller) values of $g$ and smaller (larger) values of $\mu$ and $\sigma$. Fortunately, it is common to record several chunks of signals within an experiment, so other auxiliary signals $x_1, \ldots, x_N$ obtained with the same instrument setup (and, therefore, the same gain $g$) can be exploited. Using the formalism presented in Section 3, we have that $\boldsymbol{\alpha} = (C, \mu, \sigma)$ and $\boldsymbol{\beta} = g$.

In what follows, we describe the results obtained when approximating the posterior distribution $p(C, \mu, \sigma, g | x_0, \mathcal{X})$ with Algorithm 1 using $R = 2$ rounds and $n = 50000$ simulations per round. Each simulation corresponds to 8 seconds of a signal sampled at $128\,\text{Hz}$, so each simulation outputs a vector of 1024 samples. The prior distributions of the parameters are independent uniform distributions defined as:

$$C \sim \mathcal{U}(10, 250) \quad \mu \sim \mathcal{U}(50, 500) \quad \sigma \sim \mathcal{U}(0, 5000) \quad g \sim \mathcal{U}(-30, +30)$$

where the intervals were chosen based on a review of the literature on neural mass models (Jansen and Rit, 1995; David and Friston, 2003; Deco et al., 2008). Note that the gain parameter $g$ is given in decibels (dB), which is a standard scale when describing amplifiers in experimental setups. We have, therefore, that $x(t) = 10^{g/10} s(t)$.

It is standard practice in likelihood-free inference to extract summary features from both simulated and observed data in order to reduce its dimensionality while describing sufficiently well the statistical behavior of the observations. In the present experiment, the summary features consist of the logarithm of the power spectral density (PSD) of each observed time series (Percival and Walden, 1993). The PSD is evaluated in 33 frequency bins between zero and $64\,\text{Hz}$ (half of the sampling rate). This leads to a setting with 4 parameters to estimate given observations defined in a 33-dimensional space.

The normalizing flows $q_{\phi_1}$ and $q_{\phi_2}$ used in our approximations are masked autoregressive flows (MAF) (Papamakarios et al., 2017) consisting of three stacked masked autoencoders (MADE) (Germain et al., 2015), each with two hidden layers of 50 units, and a standard normal base distribution as input to the normalizing flow. This choice of architecture provides sufficiently flexible functions capable of approximating complex posterior distributions. We refer the reader to Papamakarios et al. (2019) for more information on the different types of normalizing flows. We fix function $f_{\phi_3}$ to be a simple averaging operation over the elements of $\mathcal{X}$, so only parameters $\phi_1$ and $\phi_2$ are learned from data.

**Results on simulated data.** We first consider a case in which the observed time series $x_0$ is simulated by the neural mass model with a particular choice of input parameters. In the lower left part of Figure 3, we display the smoothed histograms of the posterior approximation $q_\phi$ obtained when conditioning on just $x_0$ ($N = 0$) or $x_0$ and $\mathcal{X}$ with $N = 9$. We see that when $N = 0$, parameters $\mu$ and $\sigma$ have large variances and that some of the pairwise joint posterior distributions have a ridge shape that reveals the previously described indeterminacy relation linking $g$ with $\mu$ and $\sigma$. When $N = 9$, the variances of the parameters decrease and we obtain a posterior distribution that is more concentrated around the true parameters generating $x_0$. This concentration is explained by the sharper estimation of the $g$ parameter, which is obtained using $x_0$ and ten auxiliary observations.

In the upper right part of Figure 3, we evaluate how HNPE concentrates around the true parameters when $N$ increases and plot the results using two other architectures: a "naive" architecture taking as context variables a stacking of $x_0$ and the elements in $\mathcal{X}$, and an "aggregation" architecture which stacks $x_0$ and the average of $\mathcal{X}$ as context variables; both architectures use a normalizing flow with 10 layers of two hidden layers and 50 units each. We evaluate the concentration of the posterior distributions via its Wasserstein distance to a Dirac located at the ground truth parameter. For each parameter in the model, i.e. $j \in \{C, \mu, \sigma, g\}$, we have three curves, $d_{\text{naive}}^j$, $d_{\text{aggreg}}^j$, and $d_{\text{HNPE}}^j$, which describe how the posterior marginal of $j$ converges to a Dirac when $N$ increases for each architecture. Since the parameters have very different scaling, we normalize the curves by dividing them by their standard value across different ground truth parameters and values of $N$. We then take the mean along $j$ so to obtain three final curves $d_{\text{naive}}$, $d_{\text{aggreg}}$, and $d_{\text{HNPE}}$. We consider ten choices of ground

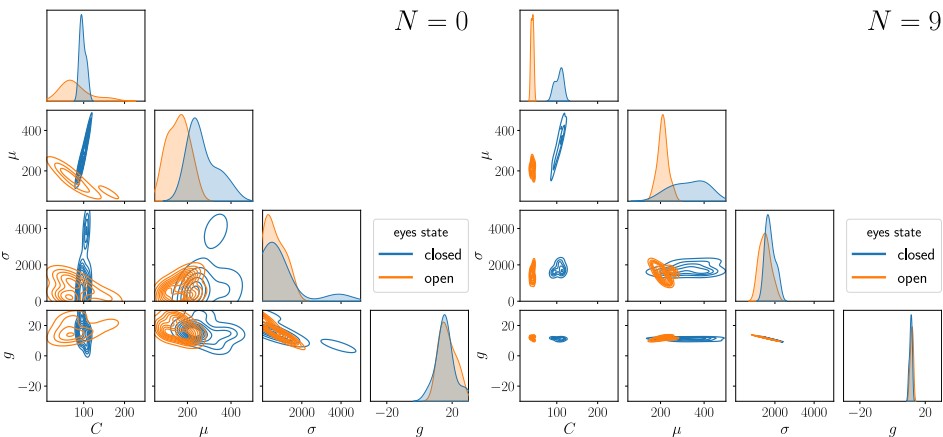

Figure 4: Posterior estimates for the parameters of the neural mass model computed on human EEG signals. Data were collected in two different experimental conditions: eyes closed (in blue) or eyes open (in orange). All signals are 8 s long and recorded at 128 Hz. We see that when $N = 9$ the posterior distributions concentrates, and that the global gain parameter gets similar in both eyes conditions. We observe that the posterior on the 3 parameters of the neural mass model clearly separate between the 2 conditions when $N = 9$.

truth parameters and show the curves with the normalized median distances. For $N = 0$, the posterior distribution is supposed to be indeterminate ("banana-shape"), so the fact that the curves do not start at the same point has no proper interpretation. For $N > 0$, the curve HNPE is uniformly below the other methods, and they converge to a *plateau*. This demonstrates the existence of a lower bound for the concentration of the posterior approximation, which can be interpreted as an irreducible variance on the estimation of the parameters. The rather good performance for the "aggregation" architecture as compared to HNPE is likely due to the fact that, for the example considered here, taking the average of the elements in $\mathcal{X}$ leads to a sufficient statistic of the observations.

We have also considered a setting in which the summary statistics of the observed time series are learned from the data instead of being fixed to the log power spectral densities, i.e. when $f_{\phi_3}$ is learned. We have used the YuleNet architecture proposed by Rodrigues and Gramfort (2020) on the example with neural mass models and report the results in Appendix B. In all our experiments, we did not see significant changes in the performance of our model so we did not include it in our evaluation as it increased the complexity of the model and its computational burden.

**Results on EEG data.** One of the most commonly observed oscillations in EEG are known as $\alpha$ waves (Lopes da Silva, 1991). These waves are characterized by their frequency around 10 Hz and are typically strengthened when closing our eyes. To relate this phenomenon to the underlying biophysical parameters of the NMM model, we estimated the posterior distribution over the 4 model parameters on EEG signals recorded during short periods of eyes open or eyes closed. Data consists of recordings taken from a public dataset (Cattan et al., 2018) in which subjects were asked to keep their eyes open or closed during periods of 8 s (sampling frequency of 128 Hz). Results for one subject of the dataset are presented in Figure 4 with $x_0$ being either a recording with eyes closed (in blue) or eyes open (in orange). We consider situations in which no extra-observations are used for the posterior approximation ($N = 0$) or when $N = 9$ additional observations from both eyes-closed and eyes-open conditions are available. When $N = 9$, we see that the gain parameter, which is global, concentrates for both eyes conditions. More interestingly, we observe that the posterior on the 3 parameters of the neural mass model clearly separate between the 2 conditions when $N = 9$. Looking at parameter $C$, we see that it concentrates around 130 for the eyes closed data while it peaks around 70 for eyes open. This finding is perfectly inline with previous analysis of the model (Jansen and Rit, 1995). Signals used in this experiment are presented in Appendix C.

## Discussion

In this work, we propose HNPE, a likelihood-free inference approach able to leverage a set of additional observations to boost the estimation of the posterior. This improvement is made possible by a hierarchical model where all available observations share certain global parameters. A dedicated neural network architecture based on normalizing flows is proposed, as opposed to the usual approach of LFI practitioners that often choose a "one size fits all" neural density estimator. We also provide a training procedure based on simulations from the model and based on the sequential approach from Greenberg et al. (2019). Although the number of additional observations ($N$) was fixed in our analysis and experiments, this parameter could be randomized and amortized during learning and enable the posterior approximation to be fed with sets of auxiliary observations of varying sizes, making it more flexible for applications. To do so, it would be necessary to simulate datasets with varying sizes of $N$ so to ensure that the several simulations are IID between them; note, however, that this would have a significant computational cost. Our approach could be extended to multi-level models using a similar factorized architecture; we did not consider such generic hierarchical models to keep the presentation clear and because our motivating examples did not require such complexity. Note, also, that HNPE could implemented with other types of conditional density estimators apart from normalizing flows, as long as the hierarchical structure of the global parameters is embedded into the structure of the approximator.

It is well known that methods for likelihood-free inference are often difficult to validate; our method is no exception. We have considered toy models for which the analytic form of the target posterior are available so to have a precise way of assessing the quality of our approximation and avoiding such difficulties. Nevertheless, further research on validation schemes for LFI methods remain of great interest, specially for more general settings for which the analytic posterior is unknown. Note, also, that LFI methods can require a large number of simulations in order to approximate the posterior distribution and may, therefore, lead to a non-negligible carbon footprint. This can be mitigated with the development of new methods for optimizing the number of simulations required for a given error tolerance, e.g. choosing the sampled parameters $\theta_i$ for which the simulations $x_i$ are the most useful for training the posterior approximation.

We demonstrated that HNPE could be reliably applied to neuroscience considering a stochastic model with non-linear differential equations. Very encouraging results on human EEG data open the door to more biologically informed descriptions and quantitative analysis of such non-invasive recordings.

## Acknowledgments and Disclosure of Funding

This work was granted access to the HPC resources of IDRIS under allocations 2021-AD011011172R1 made by GENCI. GL is recipient of the ULiège - NRB Chair on Big Data and is thankful for the support of the NRB. AG thanks the support of the ERC-StG SLAB (ID:676943) and the ANR BrAIN (ANR-20-CHIA0016) grants.

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
