# A Derivations of the posterior distributions for the motivating example

## A.1 Single observation

From Bayes' rule we have that

$$p(\alpha, \beta | x_0) \propto p(x_0 | \alpha, \beta) p(\alpha, \beta) . \tag{7}$$

Since $\epsilon$ is Gaussian we can write

$$p(x_0 | \alpha, \beta) = \frac{1}{\sqrt{2\pi\sigma^2}} \exp\left(\frac{-(x_0 - \alpha\beta)^2}{2\sigma^2}\right) , \tag{8}$$

so that the posterior is

$$p(\alpha, \beta | x_0) \propto \frac{e^{-(x_0 - \alpha\beta)^2/2\sigma^2}}{\sqrt{2\pi\sigma^2}} \mathbf{1}_{[0,1]}(\alpha) \mathbf{1}_{[0,1]}(\beta) . \tag{9}$$

We obtain an approximation to the normalization constant of $p(\alpha, \beta | x_0)$ by taking $\sigma \to 0$ and noticing that this makes the Gaussian converge to a Dirac distribution,

$$
\begin{aligned}
Z(x_0) &= \int_0^1 \int_0^1 \frac{e^{-(x_0 - \alpha\beta)^2/2\sigma^2}}{\sqrt{2\pi\sigma^2}} \mathrm{d}\alpha\mathrm{d}\beta , \\
&\approx \int_0^1 \int_0^1 \delta(x_0 - \alpha\beta) \mathrm{d}\alpha\mathrm{d}\beta .
\end{aligned}
$$

Doing a change of variables with $\gamma = \alpha\beta$ the integral becomes

$$Z(x_0) \approx \int_0^1 \left[ \int_0^\beta \delta(x_0 - \gamma) \frac{\mathrm{d}\gamma}{\beta} \right] \mathrm{d}\beta , \tag{10}$$

$$\approx \int_0^1 \frac{1}{\beta} \mathbf{1}_{[x_0, 1]}(\beta) \, \mathrm{d}\beta = \left[ \log(\beta) \right]_{x_0}^1 , \tag{11}$$

$$\approx \log(1/x_0) . \tag{12}$$

The joint posterior distribution is, therefore,

$$p(\alpha, \beta | x_0) \approx \frac{e^{-(x_0 - \alpha\beta)^2/2\sigma^2}}{\log(1/x_0)} \frac{\mathbf{1}_{[0,1]}(\alpha) \mathbf{1}_{[0,1]}(\beta)}{\sqrt{2\pi\sigma^2}} . \tag{13}$$

The marginal posterior distributions are calculated also using the fact that $\sigma \to 0$,

$$p(\alpha | x_0) = \int p(\alpha, \beta | x_0) \mathrm{d}\beta , \tag{14}$$

$$\approx \frac{\mathbf{1}_{[0,1]}(\alpha)}{\log(1/x_0)} \int_0^1 \delta(x_0 - \alpha\beta) \mathrm{d}\beta , \tag{15}$$

$$\approx \frac{1}{\log(1/x_0)} \frac{\mathbf{1}_{[x_0, 1]}(\alpha)}{\alpha} , \tag{16}$$

$$p(\beta | x_0) \approx \frac{1}{\log(1/x_0)} \frac{\mathbf{1}_{[x_0, 1]}(\beta)}{\beta} . \tag{17}$$

## A.2 Multiple observations

Suppose now that we have a set of $N$ observations $x_1, \ldots, x_N$ which all share the same $\beta$ as $x_0$ but each have a different $\alpha_i$, i.e., $x_i = \alpha_i \beta$ for $i = 1, \ldots, N$ (we consider $\sigma \to 0$ and, therefore, $\varepsilon = 0$). Our goal is to use this auxiliary information to obtain a posterior distribution which is sharper around the parameters generating $x_0$. We have that for $\mathcal{X} = \{x_1, \ldots, x_N\}$ the posterior may be factorized as

$$p(\alpha, \beta | x_0, \mathcal{X}) = p(\alpha | \beta, x_0) p(\beta | x_0, \mathcal{X}) . \tag{18}$$

Using Bayes' rule twice to rewrite the second term, we have

$$p(\beta|x_0, \mathcal{X}) \quad \propto \quad p(x_0, \mathcal{X}|\beta)p(\beta) , \tag{19}$$

$$\propto \quad \prod_{i=0}^{N} p(x_i|\beta) \, \mathbf{1}_{[0,1]}(\beta) , \tag{20}$$

$$\propto \quad \prod_{i=0}^{N} p(\beta|x_i)\mathbf{1}_{[0,1]}(\beta) . \tag{21}$$

Therefore,

$$p(\alpha, \beta|x_0, \mathcal{X}) \quad \propto \quad p(\alpha|\beta, x_0) \prod_{i=0}^{N} p(\beta|x_i) , \tag{22}$$

$$\propto \quad p(\alpha, \beta|x_0) \prod_{i=1}^{N} p(\beta|x_i) , \tag{23}$$

Using expressions (13) and (17) we obtain

$$p(\alpha, \beta|x_0, \mathcal{X}) \propto \frac{\delta(x_0 - \alpha\beta)\mathbf{1}_{[x_0,1]}(\alpha)\mathbf{1}_{[x_0,1]}(\beta) \prod_{i=1}^{N} \mathbf{1}_{[x_i,1]}(\beta)}{\log(1/x_0) \prod_{i=1}^{N} (\log(1/x_i)\beta)} . \tag{24}$$

which can be simplified to

$$p(\alpha, \beta|x_0, \mathcal{X}) \propto \frac{\delta(x_0 - \alpha\beta)\mathbf{1}_{[x_0,1]}(\alpha)\mathbf{1}_{[\mu,1]}(\beta)}{\prod_{i=0}^{N} \log(1/x_i)\beta^n} , \tag{25}$$

where $\mu = \max(\{x_0\} \cup \mathcal{X})$. The normalization constant is

$$
\begin{aligned}
Z(x_o, \mathcal{X}) \quad &= \quad \iint p(\alpha|\beta, x_0) \prod_{i=0}^{N} p(\beta|x_i) \, \mathrm{d}\alpha\mathrm{d}\beta , \\
&= \quad \int \left( \int p(\alpha|\beta, x_0)\mathrm{d}\alpha \right) \prod_{i=0}^{N} p(\beta|x_i)\mathrm{d}\beta , \\
&= \quad \int \prod_{i=0}^{N} p(\beta|x_i)\mathrm{d}\beta , \\
&= \quad \int \frac{\mathbf{1}_{[\mu,1]}(\beta)}{\prod_{i=0}^{N} \log(1/x_i)\beta^{N+1}}\mathrm{d}\beta , \\
&= \quad \frac{1}{\prod_{i=0}^{N} \log(1/x_i)} \left[ \frac{-1}{N\beta^N} \right]_{\mu}^{1} \\
&= \quad \frac{(1/\mu^N - 1)}{N \prod_{i=0}^{N} \log(1/x_i)}
\end{aligned}
$$

Then, finally, we obtain

$$p(\alpha, \beta|x_0, \mathcal{X}) = \frac{\delta(x_0 - \alpha\beta)\mathbf{1}_{[0,1]}(\alpha)\mathbf{1}_{[\mu,1]}(\beta)}{(1/\mu^N - 1)} \frac{N}{\beta^N} . \tag{26}$$

Simple integrations show that

$$p(\alpha|x_0, \mathcal{X}) = \frac{\mathbf{1}_{[x_0,\min(1,\frac{x_0}{\mu})]}(\alpha)N\alpha^{N-1}}{(1/\mu^N - 1) \, x_0^N} \tag{27}$$

$$p(\beta|x_0, \mathcal{X}) = \frac{\mathbf{1}_{[\mu,1]}(\beta)N}{(1/\mu^N - 1) \, \beta^{N+1}} \tag{28}$$

# B The neural mass model

## B.1 A cortical column as a system of stochastic differential equations

The neural mass model used in our work is the one presented in Ableidinger et al. (2017). This is an extension of the classic Jansen-Rit model (Jansen and Rit, 1995) to make it compatible with a framework based on stochastic differential equations. The model describes the interactions between excitatory and inhibitory interneurons in a cortical column of the brain. In mathematical terms, the model consists of three coupled nonlinear stochastic differential equations of second order, which can be rewritten as a six-dimensional first-order stochastic differential system:

$$
\begin{aligned}
\dot{X}_0(t) &= X_3(t) \\
\dot{X}_1(t) &= X_4(t) \\
\dot{X}_2(t) &= X_5(t) \\
\dot{X}_3(t) &= \left( Aa\big(\mu_3 + \mathrm{Sigm}(X_1(t) - X_2(t)) - 2aX_3(t) - a^2 X_0(t)\big) \right) + \sigma_3 \dot{W}_3(t) \\
\dot{X}_4(t) &= \left( Aa\big(\mu_4 + C_2\, \mathrm{Sigm}(C_1 X_0(t)) - 2aX_4(t) - a^2 X_1(t)\big) \right) + \sigma_4 \dot{W}_4(t) \\
\dot{X}_5(t) &= \left( Bb\big(\mu_5 + C_4\, \mathrm{Sigm}(C_3 X_0(t)) - 2bX_4(t) - b^2 X_2(t)\big) \right) + \sigma_5 \dot{W}_5(t)
\end{aligned}
\tag{29}
$$

The actual signal that we observe using a EEG recording system is then $X(t) = 10^{g/10}(X_1(t) - X_2(t))$, where $g$ is a gain factor expressed in decibels. According to Jansen and Rit (1995), most physiological parameters in (29) are expected to be approximately constant between different individuals at different experimental conditions, except for the connectivity parameters $(C_1, C_2, C_3, C'4)$ and the statistical parameters of the input signal from neighboring cortical columns, modeled by $\mu_4$ and $\sigma_4$. Following the setup proposed in Buckwar et al. (2019), we then define our inference problem as that of estimating the parameter vector $\boldsymbol{\theta} = (C, \mu, \sigma, g)$ from an observation $X_{\boldsymbol{\theta}}$, where $\mu = \mu_4$ and $\sigma = \sigma_4$, and the $C_i$ parameters are all related via $C_1 = C, C_2 = 0.8C, C_3 = 0.25, C_4 = 0.25C$.

## B.2 Choice of summary statistics

The inference procedure is then carried out not on the time series itself but on a vector of summary statistics. The results described in Section 4.2 were obtained with a fixed choice on the power spectral density of the time series as summary statistics. However, it is possible (and very often preferable) to learn the best summary statistics from data. We have considered this option using the YuleNet proposed in Rodrigues and Gramfort (2020), where a convolutional neural network is jointly learned with the approximation to the posterior distribution. Figure 5 portrays the results obtained with different numbers of auxiliary observations in $\mathcal{X}$. Note that the 'quality' of the approximation seems to stagnate when $N > 10$ as observed also in Figure 3. We did not carry out more experiments on this data-driven setting because of difficulties due to numerical instabilities in the training procedure when $N$ increases and for certain choices of ground truth parameters. Also, the memory consumption using YuleNet with large values of $N$ makes the use of GPU a challenge. We intend to continue investigations with learned summary statistics in future works.

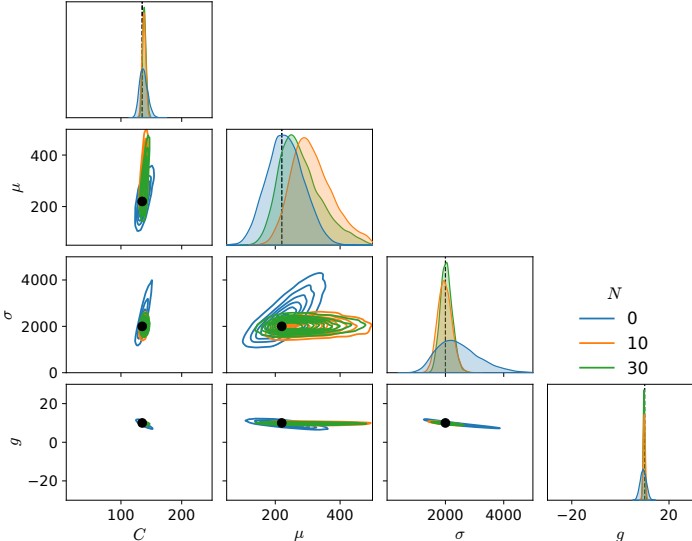

Figure 5: Posterior estimates for the parameters of the neural mass model obtained on 8 s of data sampled at 128 Hz and simulated using $C = 135$, $\mu = 220$, $\sigma = 2000$, and $g = 10$. One can observe that increasing $N$ allows to concentrate the posterior on the correct parameters.

## C  EEG data

The EEG signals used for generating the results in Figure 4 are displayed in Figure 6. We have used only the recordings from channel Oz because it is placed near the visual cortex and, therefore, is the most relevant channel for the analysis of the open and closed eyes conditions. The signals were filtered between 3 Hz and 40 Hz.

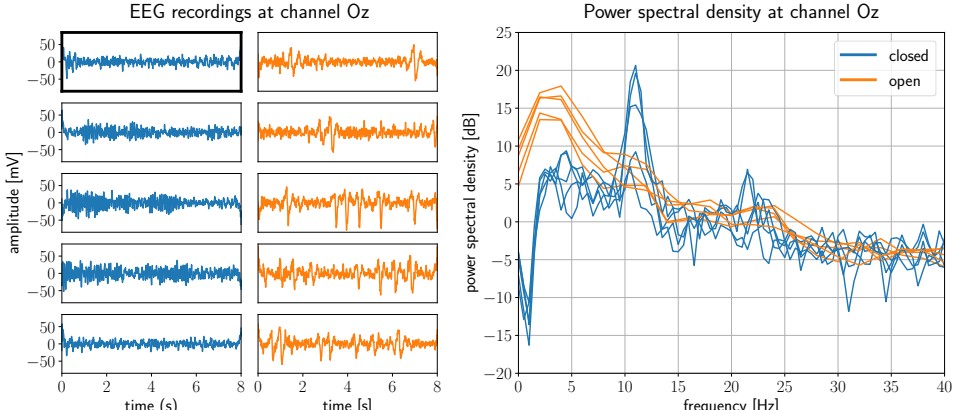

Figure 6: EEG data used on our analysis described in Figure 4. (**Left**) All ten time series considered in our analysis. The plot with thicker bounding boxes is the observed signal $x_0$ in the closed eyes state. All other time series belong to $\mathcal{X}$. (**Right**) Power spectral density of each time series calculated over 33 frequency bins. These are the actual summary features used as input in the approximation of the posterior distribution.