# OpenReview forum: "HNPE: Leveraging Global Parameters for Neural Posterior Estimation"
_NeurIPS.cc/2021/Conference — NeurIPS 2021 Poster_

### Official Review · Reviewer_u4up · 2021-07-12

**Rating:** 6
**Confidence:** 3

**Summary:**

The paper addresses the inference of illposed models, where the likelihood is intractable, and proposes to utilize hierarchical structures in combination with neural density estimators (normalizing flows) to ameliorate identifiability of the parameters. It motivates the approach with a simple analytical tractable example, and validates the approach on modelled and real EEG time-series.

**Limitations And Societal Impact:**

+ As mentioned above, I’d like to see some additional and rigour discussion on when such separation of variables helps the inference.
+ Please report number of parameters of models, what device was used to train them etc. to allow the reader to evaluate the showed results.
+ In general, more experiments, that demonstrate the improvement achieved by the proposed method
+ I sometimes feel, the paper is written too complicated. For example, in l134 it is written that $f_{\phi^{(3)}}$ is “invariant to the ordering of the input”, which is obvious, since the function in question is a sum over inputs (which is commutative). I thought quite a bit about this comment, and I am not sure whether I am not seeing something, or why it is important to mention here. Another example is, that $f_{\phi^{(3)}}$ seems never to be used in the experiments, which makes me wonder, why is it introduced in the first place.
+ Some typos:
++ L129f: indices of phi are mixed
++ There are several approximation signs in the appendix that should be equal, and some equal signs that should be approximations.
++ L107: It should be \beta not \beta_0


**Main Review:**

The problem the authors consider is interesting and important for scenarios, where models are parametrized ambiguously. However, I consider the present work as too immature for publication, because to me, the proposed approach (making use of additional observations and separating the parameters in global and local ones) is in my opinion not sufficiently justified explored. Some questions I would like to see answered are
+ When is the separation off variables into local and global possible or desirable. E.g. when you remove the assumption that alpha and beta are upper bounded by 1 in the motivating example, the separation wouldn’t help any more in my opinion. I do not say, that the motivating example isn’t interesting, but I would like to have seen more discussion, on when it makes sense to have this separation. Hence, I would appreciate some general discussion, when such the proposed separation makes sense.
+ The problems, that are considered, are very low dimensional, which is making me wonder whether the neural density estimators are really necessary here. Just demonstrating that the separation ameliorates the inference by using Gaussians (or mixtures of them, or other simple estimators) for the density approximation in Eq. (5), might be more insightful, and easier to train.
+ Overall, I was missing details about how many parameters are used in the different models, how long it takes them to train, etc. So I couldn’t make myself a picture, whether comparisons are fair or not.
+ For the fact, that the paper doesn’t provide any theory (except for a specific toy example), how this separation is helping model inference, I think the empirical results are too little, to consider the proposed method as helpful in general.

### Update

Reading the comments of the other reviewers, and the response of the authors, I am reconsidered my score and raised it to 6 (though decreasing my confidence score). This is because,

+ Reading the reviewers' comments, it seems I underestimated the contribution of the paper, leveraging information of the hierarchical structure of models in the context of likelihood-free inference.
+ I mentioned originally, that I found the manuscript written too complicated and mentioned as examples invariance of the averaging operation to permutations, and that $\phi_3$ is introduced, but seems to be never learnt. I understand for the first point, that though trivial, it is a crucial point for the construction of the neural posterior, so it makes sense to give the fact more space than needed on first sight. The fact, that $\phi_3$ is not needed in the shown experiments (as I got from the response to reviewer fnEF), still poses the question, whether it is necessary to introduce it, or just to mention the possibility of having, but simplify notation.
+ Experiments seem to be computational heavy to provide new baselines for the NMM example, so it is maybe too much to ask.

**Time Spent Reviewing:**

5h

---

> ### Author Response · Authors · 2021-08-09
> **Further precisions on the justification of our proposal**
>
> **Intro**
>
> We thank the reviewer for appreciating the relevance of the problem that we consider in our paper. We answer below the questions and concerns raised by the reviewer.
>
> **Answer to question on the separation of variables**
> > “When is the separation of variables into local and global possible or desirable”
>
> The separation of variables into local-global is inherent to the kind of problem that we have considered in this paper, which are Bayesian hierarchical models. There are several applied settings where such models appear, such as bioinformatics, population studies, and natural language processing. Indeed, there is a vast literature on the topic and entire books dedicated to it (see e.g, Congdon 2010, for an introductory book on Bayesian hierarchical models); the neuroscience example that we have considered is one of such models. With that said, identifying whether we can separate parameters into local and global parameters (and which ones to consider in each category) is more of a choice of the modeller.
>
> **Answer to remark on the low-dimensionality of the problem**
> > “The problems that are considered are very low dimensional, which is making me wonder whether the neural density estimators are really necessary here”.
>
> Likelihood-free inference methods are often applied to relatively low-dimensional settings and, nevertheless, posterior approximations using neural density estimators have been demonstrated to be superior as compared to simpler approximators in several papers. See for example [Papamakarios and Murray 2016] where LFI was carried out with mixture density networks and more recent papers such as [Papamakarios et al. 2018] and [Greenberg et al. 2019] where superior results are obtained with normalizing flows.
>
> **Answer to question on the architectures of the neural networks**
> >“Overall, I was missing details about how many parameters are used in the different models, how long it takes them to train, etc. So I couldn’t make myself a picture, whether comparisons are fair or not.”
>
> Likelihood-free inference relies on a training dataset consisting of several simulations from the model under study. These simulations are, in general, its main computational bottleneck and we have reported the number of simulations used in each comparison exactly to take this fact into account. We have given details of the number of training parameters, the architecture of the flows, etc. in the text (lines 185 and 276). Following the suggestions by other reviewers, we have improved the fairness of the comparisons of h-Flow versus the naive architecture by making them have the same number of MAF layers (and, therefore, the same capacity); we describe these changes in our response to Reviewer 2 (2NmZ).
>
> **Answer to remark on the lack of empirical results**
> >“I think the empirical results are too little to consider the proposed method as helpful in general.”
>
> This is a rather subjective and poorly grounded opinion. We would appreciate it if the reviewer gave a more factual assessment of our work and was more precise on why he thinks our empirical assessment is not sufficient. We invite the Reviewer to read our response to Reviewer 1 (tXAy), where we address concerns regarding the thoroughness of our experiments.
>
> **Answer to remark on the complexity of the text**
> > “I sometimes feel the paper is written too complicated. [...] it is written that $f_{\phi_3}$ is   ‘invariant to the ordering of the input’, which is obvious, since the function in question is a sum over inputs (which is commutative)”
>
> We agree that the commutative property of $f_{\phi_3}$ is quite evident from its definition, but we believe it is important to stress such invariance because it is not systematically discussed (or taken into account) in other works from the likelihood-free inference literature. If one ignores such invariance and skips the aggregation phase that we propose, the resulting posterior approximator is much worse, as we show with the naive architecture. Also, we would appreciate it if the reviewer could point us to which aspects of our text seem too complicated as this would help us improve our presentation.
>
> **Conclusion**
>
> In light of our effort to clarify all of the reviewer’s questions and concerns, we hope that he/she will improve our score if he/she is satisfied by them.

---

### Official Review · Reviewer_fnEF · 2021-07-14

**Rating:** 6
**Confidence:** 4

**Summary:**

This paper proposes an approach to perform statistical inference for hierarchical models with intractable likelihoods. It extends an existing approach for simulation-based inference (SNPE-C) in order to use models that have global and local random variables.

The authors propose a scheme in which two conditional flows a trained: The first flow models a distribution over global parameters given an observation and $N$ permutation invariant auxiliary observations. Permutation invariance could e.g. be accounted for with a deepset architecture. The experiments do not make use of this flexibility, relying on an average operation instead. The second flow then models a distribution over local parameters given the observation and global parameters. They refer to the resulting inference scheme jointly as h-Flow, for hierarchical flow.

A motivating toy example is provided, along with a relevant, more realistic simulator from Neuroscience (a Neural Mass Model; NMM). h-Flow is compared against a naive approach on both of these examples. The comparison turns out about equal on the toy example and favorably for h-Flow on the NMM. On the toy example, h-Flow is additionally compared against a hierarchical ABC variant and LFVI, comparing favorably to both of them.

**Limitations And Societal Impact:**

The authors state in their checklist that potential negative societal impacts of their work are not applicable. I agree that an extended discussion of negative impacts might not be strictly necessary, given the nature of this work is about statistical inference applied to a certain class of models (with hierarchical structure and intractable likelihoods). Thinking more broadly, the societal impact and potentially adverse effects will depend on the setting in which the algorithm gets deployed. However, I would suggest to at least briefly touch upon the general difficulties associated with diagnostics/criticism in the SBI context.

**Main Review:**

# Main Review

## Significance and Originality

Overall, the paper is well motivated and introduces a strategy which can potentially be useful for many applications of SBI. The proposed solution might come at no big surprise, but in fact this can be viewed as a strength: As long as the proposed strategy of training two flows (see summary above) performs robustly and demonstrably better than what one would naively do with SNPE-C, and given that it outperforms existing methods, I would view this paper as being a valuable addition to the conference that should be presented.


## Clarity

The paper is generally well-written and easy to follow. I especially enjoyed that the authors introduced a motivating example that then finds its structure reflected in the applied setting.


## Quality

My reasons for not giving a higher score, and currently tending to rather reject the paper, are many open questions and some specific requests I have regarding the comparisons. I detail these concerns below and hope that the authors will address them, in which case I will revise my evaluation accordingly.

### Comparisons to Naive Approaches

My primary concern is that the paper does not yet clearly demonstrate that the proposed method is better than naive approaches using SNPE-C with a single flow (instead of using two flows, as h-Flow does). In the toy experiment, no difference between h-Flow and the naive approach is found (Figure 2). On the NMM, a difference is clearly visible in Figure 3, but as of now I am unsure how to interpret this result exactly and how it was obtained. Some specific questions and requests that can hopefully help clarify this issue follow.

**Q1**: In Figure 3, the difference between the naive approach and h-Flow _is biggest_ for $N=0$. I was surprised by this, since my understanding is that we expect improvements in resolving indeterminacies by additional observations, i.e. when $N>0$. In the toy example, which served as a motivation for the NMM, as $N \rightarrow 0$ the posterior distribution converges to $p(\alpha, \beta | x_0)$. In other words, for the toy example, we do not expect a difference (and empirically don't find) a difference in performance when $N=0$. Could the authors comment on the difference in performance for $N=0$ on the NMM?

**Q2**: Regarding the results presented in Figure 3 the authors write: "We see that the factorization scheme yields uniformly better results than the naive architecture. This is mainly because the dimensionality of the context variables in the naive case gets too big when more observations become available, making the training of the normalizing flow in the non-factorized setting harder.“. Why exactly do the context variables in the naive case get too big with more observations? I would have thought that for the naive scheme we simply aggregate the additional observations before passing them as context to the single flow. If we do not aggregate them, we assume that the ordering matters, which is would put the naive implementation at a big disadvantage and is not how one would naively apply SNPE-C with a single flow in this context. It would be important to comment and clarify this.

**R1**: As a request for new experiments: The results in Figure 3 are reported in terms of Wasserstein distance between a dirac delta on ground truth parameters and the inferred posterior (median without error bars for 5 repeats). It would be important to extend this analysis for a much larger number of ground truth parameters and report results such that the variability can be evaluated (e.g. mean and standard errors). Ideally, a secondary metric to the Wasserstein distance was reported in the supplement, such as the more commonly used negative log probability of true parameters _averaged_ over as many ground truth parameters as feasible (this is crucial, otherwise this metric would be hard to interpret).

### Results for h-ABC and LFVI on NMM

**R2**: As a second request, it would be good to provide results for h-ABC and LFVI on the NMM model for completeness.


## Additional questions

**Q3**: In Figure 1 (right) we observe that the $N=10$ case is worse relative to $N=0$. To me this seemed counterintuitive at first and I think it would be good if the authors commented on it in the manuscript. My interpretation is that the density estimation problem gets more difficult when we have more observations (posterior gets more concentrated). So resolving indeterminacies leads to less accurate inference (as compared to the analytic solution). Would the authors agree with this interpretation?

**Q4**: "We did not carry out more experiments on this data-driven setting because of difficulties due to numerical instabilities in the training procedure when N increases and for certain choices of ground truth parameters.". It would be important to investigate this issue further, if possible for the current publication. Why would this only occur for certain ground truth parameters and not for others?

**Q5**: For the toy example the authors set $\sigma=0$, so that $\epsilon = 0$. Usually SBI is only applied in the context of stochastic simulators. Would results change in a meaningful way if the experiments were repeated for a small value of $\sigma$?

**Q6**: The NSF in Figure 1 does not seem able to handle the sharp transitions that occur when $N=10$ or $N=100$. Do the authors have any suggestions to remedy this?

**Q7**: On the NMM, MAFs instead of the more expressive NSFs are used (whereas the toy example does use NSFs). What are the reasons behind this choice and how do results change for h-Flow and the naive strategy when using them instead?

**Q8**: "Although the number of additional observations (N) was fixed in our analysis and experiments, this parameter could be randomized and amortized during learning and enable the posterior approximation to be fed with sets of auxiliary observations of varying sizes making it more flexible for applications.“ $\rightarrow$ My understanding is that the aggregation is simply done through the average in the experiments. What exactly would not work/require changes to have flexible N?

**Q9**: Have the authors done any internal experiments in which $\phi_3$ was learned as well? If so, were there any problems that they encountered that restricted the experiments to learning $\phi_1$ and $\phi_2$ only?


## Name of the algorithm

I see two potential disadvantages associated with with naming the algorithm h-Flow. They are:
1. One of the strengths of SNPE is that it can be used with any conditional density estimator. In the future, perhaps entirely different classes of CDEs will be invented. I might be misunderstanding something (please correct me if I'm wrong) but to me it seems there is nothing flow-specific about the proposed algorithm. In particular I assume that we can use it with all kinds of CDEs as long as they work with embeddings/can be designed to be permutation invariant and we can evaluate log probabilities and draw samples. For example, we could just as well use this with MDNs, which might be more suitable depending on the application/knowledge we have about a problem.
2. h-Flow in the paper title sounded to me as if a single architecture was trained, i.e. that this paper was about an entirely novel flow architecture. Instead, two flows are trained, and if the approach gets extended to multi-level architectures, training would involve more and more flows, not a single one.

**Q10**: What are the authors thoughts on these remarks regarding their algorithm's name?

## Minor

- Fig. 3: There are display issues with this figure when opening the PDF using Preview on macOS. Most marginals are missing and the x-axis of the inset figure is cut off (3 instead of 30)
- L11: validate quantitatively our proposal $\rightarrow$ validate our proposal quantitatively
- L103: have to be approximated $\rightarrow$ we decide to approximate them
- L178: $5 \times 10^{−4}$ rather than $5.10^{−4}$?
- L227: captures well the global parameter $\rightarrow$ captures the global parameter well
- L229: could well estimate local parameters $\rightarrow$ could estimate local parameters well

## Update

I thank the authors for their reply and have updated my score.

**Time Spent Reviewing:**

7

---

> ### Author Response · Authors · 2021-08-09
> **About our comparisons against the naive approach**
>
> **Intro**
>
> We thank the reviewer for appreciating the relevance of our work and the clarity of the manuscript, as well as for saying that our work would be a valuable contribution to NeurIPS 2021. We are also particularly grateful for the very thorough review and the important remarks/suggestions.
>
> In the next few paragraphs, we answer each one of the questions of the reviewer.
>
> **Answer to Q1**
> > “Could the authors comment on the difference in performance for $N$=0 on the NMM?”
>
> The metric shown in the figure is the Wasserstein distance between the posterior distribution and a Dirac at the ground truth. This quantity measures both how the posterior is biased with respect to the true parameter value and the variance around its mean (one can show analytically that it consists of the sum of these two quantities). When $N=0$, we expect the posterior distribution to have a “banana-shape” reflecting the ill-posedness of the model. It is, therefore, normal to observe high values of bias+variance, since these concepts make no sense for a multimodal posterior distribution. This explains why we observe high values both for $h$-Flow and its naive counterpart; the fact that one is larger than the other is probably related to differences in the capacity of each architecture and do not necessarily reflect that one is better than the other. However, once N starts increasing, we expect the posterior distribution to concentrate around the ground truth parameter and we see that $h$-Flow concentrates faster than the naive implementation. This is an important observation that we shall include in the camera-ready version of our paper.
>
> Please note that we have rerun our experiments to try ensuring similar capacities between $h$-Flow and the naive implementation and, therefore, a fairer comparison between them: the naive architecture has now 6 MAF layers and the $h$-Flow has two normalizing flows (one for each factor) with each one composed of 3 MAF layers. Our new results are very similar to those from our initial submission and still show differences between the results for each architecture when $N=0$. A new figure will be included in the camera ready version of our paper to show these results.
>
> **Answer to Q2**
> > “Why exactly do the context variables in the naive case get too big with more observations?”
>
> Indeed, our initial implementation of the naive architecture does not aggregate the extra observations from $\mathcal{X}$ and, therefore, has the effect of increasing the dimensionality of the context variables. We act as such because the development of strategies for making neural networks permutation invariant is rather recent in the literature and not necessarily known to all, so it didn’t seem too unrealistic to consider such architecture. We understand, however, the concerns raised by the reviewer saying that this implementation would be “too naive” and have a great disadvantage with respect to $h$-Flow. This is why we have rerun our experiments for NMM with a naive architecture that actually aggregates the extra observations (on top of the changes in the capacity of the flow). The new curves have the same qualitative behavior as before and indicate that the factorized approach (i.e. $h$-Flow) concentrates faster around the ground truth parameter (i.e. with fewer extra observations) than the naive one; we will include these new results in our camera-ready version of the paper.
>
> **Answer to R1**
> > “It would be important to extend this analysis for a much larger number of ground truth parameters and report results such that the variability can be evaluated”
>
> We thank the reviewer for this suggestion. We will add results with more ground truth parameters in the camera-ready version of our paper. This will indeed allow for a more complete picture of the variability of our results.
>
> **Answer to R2**
> > “As a second request, it would be good to provide results for h-ABC and LFVI on the NMM model for completeness.”
>
> We have deliberately chosen to consider only $h$-Flow and its naive counterpart for the NMM because the h-ABC and LFVI had already demonstrated rather poor performance on the toy model. It did not seem to us very reasonable to simply run more simulations and spend more energy over weeks of computation on two clearly inferior methods for the sake of completeness. However, we are open to run such experiments if the reviewer believes it is a major point to include.
>
> **Answer to Q3**
> > “My interpretation is that the density estimation problem gets more difficult when we have more observations (posterior gets more concentrated). So resolving indeterminacies leads to less accurate inference (as compared to the analytic solution). Would the authors agree with this interpretation?”
>
> Yes, we agree with the interpretation of the reviewer. We will add a sentence in the camera-ready version of our paper clarifying this important point.
>
> **Answer to Q4**
> > “Why would this only occur for certain ground truth parameters and not for others?”
>
> The neural mass model is known to have rather unstable behaviour on certain regions of the parameter space (e.g. exploding amplitudes with no physiological meaning, lack of oscillations, etc.), which is why we can expect that the inference for certain ground truth parameters could be harder than for others.
>
> **Answer to Q5**
> > “Would results change in a meaningful way if the experiments were repeated for a small value of $\sigma$?”
>
> We have considered $\sigma = 0$ in our experiments because it allows us to write analytic expressions for the posterior distribution and have theoretical insights of the results expected during the inference. All the important qualitative results obtained for $\sigma = 0$ are the same for small values of $\sigma$, whereas a $\sigma$ that is too large will make the inference simply impossible (low signal to noise ratio).
>
> **Answer to Q6**
> > “The NSF in Figure 1 does not seem able to handle the sharp transitions that occur when $N=10$ or $N=100$. Do the authors have any suggestions to remedy this?”
>
> One possibility would be to consider different base distributions for the normalizing flow instead of the usual Gaussian distribution. We have not tried such a setting, but there have been a few other works in the literature [Jaini et al. 2020 on arXiv:1907.04481] that have managed to improve the tail-behavior of probability densities approximated by flows when using other base distributions (e.g. t-student).
>
> **Answer to Q7**
> > “On the NMM, MAFs instead of the more expressive NSFs are used (whereas the toy example does use NSFs). What are the reasons behind this choice?”
>
> We did try both architectures but have observed better results with NSF in the toy models whereas MAFs lead to better results in the NMM example. One hypothesis is that the NSF architecture better captures the discontinuities of the posterior distribution of our toy model as compared to MAF and is well adapted for the case of univariate distributions, whereas the MAF is more robust and easier to train when handling multivariate settings such as in the NMM example.
>
> **Answer to Q8**
> >“My understanding is that the aggregation is simply done through the average in the experiments. What exactly would not work/require changes to have flexible $N$?”
>
> It would be necessary to simulate datasets with varying sizes of $N$ so to ensure that the several simulations are IID between them.
>
> **Answer to Q9**
> >“Have the authors done any internal experiments in which $\phi_3$ was learned as well?”
>
> Yes we did a few experiments where $\phi_3$ was learned in the Toy model example. We did not manage to find good heuristics or theoretical improvements as compared to taking simply the average values of the observations. We intend to investigate this question in the future.
>
> **Answer to Q10**
> >“What are the author's thoughts on these remarks regarding their algorithm's name?”
>
> We thank the reviewer for this important remark. We agree that the name of our method does not reflect the generality of our approach and gives the impression that one can only apply these ideas to settings where normalizing flows are used. With that said, in the camera-ready version of the paper we intend to change the name of our method to “HNPE” which stands for hierarchical neural posterior estimation.
>
> **Answer to remark on limitations and societal impact:**
> > “I would suggest at least briefly touch upon the general difficulties associated with diagnostics/criticism in the SBI context.”
>
> We thank the reviewer for this important suggestion. We will include a new sentence regarding the difficulties of diagnosing the quality of the results from SBI and, therefore, being confident of what can be concluded from them. Also on this topic, we shall mention that SBI is, by definition, based on the idea of doing several computer simulations. It is, therefore, an intrinsically computationally expensive method that could raise issues regarding its possible environmental impact. For instance, is it always justified to do more and more experiments to validate a new method or choose well thought examples that serve as sufficient proof of good performance.
>
> **Conclusion**
>
> In light of our responses to all of the reviewer’s questions and the new experiments that have been carried out following his suggestions, we hope the reviewer will improve his/her score if he/she is satisfied with our efforts and convinced by our responses.

---

### Official Review · Reviewer_2NmZ · 2021-07-15

**Rating:** 6
**Confidence:** 3

**Summary:**

The authors propose a new likelihood-free inference method for Bayesian hierarchical models, called h-Flow, that exploits additional information provided by a set of auxiliary observations with shared, global parameters. h-Flow works by approximating two relevant posterior distributions, i.e. that of the global parameter beta and the local parameters alpha given beta, with separate normalizing flows, which are then multiplied to yield the joint posterior distribution. The normalizing flows are trained by minimising the KL-divergence between the true and approximated posterior distribution and the training data is refined over several rounds by means of SNPE-C (also known as APT).

**Limitations And Societal Impact:**

The authors did not comment on limitations of their method and the potential societal impact it may have.


**Main Review:**

The authors provide an approach to likelihood-free inference, which is a known and important task, for Bayesian hierarchical models. While the method is new, to me it appears to be a slight variation of SNPE-C that includes a second normalizing flow for global parameters that is conditioned on a set of auxiliary variables. I am thus mildly worried about novelty and significance. I would encourage the authors to emphasise their contributions and the difference to related work further.

The derivations are generally clear and technically sound. While the examples in their experiments are very interesting, I don't think they highlight the method very well. It is still unclear to me whether or not any improvements are due to the proposed method, or a) an increased flexibility or b) simply having more auxiliary observations that are correlated with new observations. The authors attempt to clarify this by using a naive approach that ignores the hierarchical structure, but this only consists of one normalizing flow with a lower capacity than the two flows of h-Flow. Similarly, as the authors said, N>0 generally yields better posteriors but this is naturally not very surprising. In my opinion, a fair baseline for e.g. the EEG experiments would have been a comparison of N=0 vs N=9 with both naive architectures and h-Flow, but a setting where both approaches have the same capacity.

The submission is generally clearly-written, at least until Section 3. I believe the experimental section could have been more concise with some explanations about the models and approaches being moved to the appendix, leaving space for, perhaps, one more experiment or more elaborate baseline comparisons.

**Time Spent Reviewing:**

3

---

> ### Author Response · Authors · 2021-08-09
> **About the novelty of $h$-Flow**
>
> **Intro**
>
> We thank the reviewer for appreciating the clarity of our manuscript and the soundness of the method that we propose.
>
> **Answers to the main remarks of the reviewer**
>
> The reviewer states that
> > While the method is new, to me it appears to be a slight variation of SNPE-C'
>
> but we would like to point out that it actually extends SNPE-C in two non-trivial ways:
> - Factorization using two normalizing flows and
> -Aggregation of observations
>
> Such improvements resulted in clear improvements for hierarchical models (cf. comparisons between $h$-Flow and naive implementations) and are made possible by exploiting the structure of the statistical model. Indeed, LFI practitioners often follow a “one size fits all” approach for normalizing flows, completely ignoring the intrinsic structure of the problem. An important contribution of our work has been to illustrate a situation where taking the structure of the simulator model is very useful for inference and ensures better results with fewer simulations.
>
> The reviewer also states that
> > It is still unclear to me whether or not any improvements are due to the proposed method, or a) an increased flexibility or b) simply having more auxiliary observations that are correlated with new observations
>
> We should clarify that the presence of more auxiliary observations is the precise reason for which the posterior distribution gets sharper around the ground truth parameters; this is a direct consequence of the concentration of $p(\beta|x_0, \mathcal{X})$ around the ground truth value of $\beta$ when $N \to \infty$. Our contribution relies on the fact that we have managed to profit from such a hierarchical structure and propose an architecture that exploits this property. Such architecture is based on two normalizing flows and we have demonstrated its superior performance as compared to an approach based on a single flow (naive implementation). We agree with the reviewer that such comparison would be fairer if we had ensured that both architectures had the same capacity. We have, therefore, run new experiments in which both architectures (h-Flow and naive) have the same number of parameters: the $h$-Flow is implemented with two normalizing flows consisting of 3 MAF layers and the naive implementation is a single normalizing flow with 6 MAF layers; all layers have the same number of parameters. The new results on the NMM are very similar to those obtained in our first submission and confirm the superiority of our factorized approach versus the naive implementation. We will include this new result in the camera-ready version of our paper.
>
> **Conclusion**
>
> In light of our responses to the reviewer’s questions and the new experiments that have been carried out to make a fairer comparison between methods, we hope the reviewer will improve his/her score if he/she is satisfied with our efforts and convinced by our new results.

---

### Official Review · Reviewer_tXAy · 2021-07-16

**Rating:** 7
**Confidence:** 3

**Summary:**

The paper extends SNPE-C to the situation where multiple sets of parameters yield identical observations, but where the parameters can be split into local and global ones, which are specific to, or shared across all observations, respectively. It is shown how this additional information can be used to build a flow-based posterior estimator yielding sharper distributions.


**Limitations And Societal Impact:**

No concerns seen here.

**Main Review:**

The paper considers simulation-based inference in the presence of strongly coupled parameters leading to a non-injective likelihood function. This parameter estimation problem for such a system is ill-posed, but under additional assumptions this can be resolved with a hierarchical Bayesian model in which some parameters are shared across all observations.

The authors propose a practical realization of this solution in the form of a "h-flow" model, consisting of two normalizing flows modeling the posterior for the local and global parameters, and utilizing a deepset architecture for the global parameters to enforce permutation invariance. The network is optimized with KL divergence, and the authors show how to train it within the multi-round SNPE-C scheme.

h-flow is then experimentally tested on a simple model dependent on the product of two parameters, and on a 4d neural-mass model (NMM) of a cortical column with 3 local parameters and 1 global one. Performance is comparable to a naive normalizing flow in the simple case, but a significant improvement is seen for the NMM. To further show practical applicability, the authors then use h-flow to estimate NMM parameters on real EEG signals recorded under eyes open/closed conditions, and show that this results in separated distributions for the two conditions, in line with expectations.

The paper is clearly written, provides simple and illustrative motivating examples, and keeps the most relevant information in the main text. The proposed method is practically useful, and easy to implement, which should lead to adoption among practitioners already using SNPE-C or similar SBI methods. The experimental results could be made a bit more convincing with additional models or more baseline methods in the case of the NMM.

----
Score revised +1 based on the authors' responses and the promise to report improved baselines in the NMM test case.

**Time Spent Reviewing:**

4

---

> ### Author Response · Authors · 2021-08-09
> **Concerning how convincing our experimental results are**
>
> **Intro**
>
> We thank the reviewer for appreciating the clarity of our manuscript and the relevance of our work for the SBI community.
>
> **Answer to main remark of the reviewer**
>
> The reviewer states that
> >The experimental results could be made a bit more convincing
>
> and we describe next why we believe that such experiments demonstrate the relevance and importance of our method in a very satisfying manner. We hope that such clarification will lead the reviewer to increase our score.
>
> Firstly, although we agree that we could have included more models in our experiments, we believe that it was more valuable to have a simple (yet not trivial) motivating example for which analytic results were available and precise quantitative analysis could be carried out. Indeed, evaluating the quality of a new likelihood-free inference (LFI) method is a difficult task in general, because of the intractability of the posterior distribution being approximated. Our choice ensured we would have theoretical insights and interpretations about our results instead of just several scalar values which are hard to interpret. It should be noted that such an approach is rather unique in the LFI literature and, to the best of our knowledge, the first one in the context of hierarchical Bayesian models.
>
> Note, also, that we have deliberately chosen to do all comparisons of our method versus previous ones from the literature (i.e. LFVI and h-ABC) on the toy model because of the possibility of doing a quantitative assessment of how far the approximated posterior distribution was from the true one. These comparisons demonstrated that both concurrent approaches perform very poorly on the toy model as compared to our proposal. Because of such results, we have decided that for the neural mass model (NMM) we would only compare methods that actually performed well on the toy model, i.e. $h$-Flow and its naive implementation using a single normalizing flow. Please note that we have taken into account the concerns raised by Reviewer 2 (2NmZ) and have now considered a new set of experiments with a fairer comparison of the naive model against $h$-Flow on both the toy model and the NMM (see our answer to Reviewer 2 (2NmZ) for more details)
>
> Finally, we would like to add that our work tackles a complete LFI pipeline, going from the theoretical model (neural mass model) all the way to real EEG recordings. Such full analysis is very rare in the literature and is an important illustration of how one can actually use simulation-based inference to link real data with theoretical models.
>
> **Conclusion**
>
> We hope that in light of our responses to the reviewer’s concerns, he/she will improve our score if he/she is convinced and satisfied by them.

---

> > ### Comment · Reviewer_tXAy · 2021-08-30
> > **Response to authors**
> >
> > Thank you for the response! I have increased my score based on your extended results covering a more comparable baseline for the NMM test case with matched network capacity and observation aggregation. I appreciate your inclusion of a test case involving real data, and it is indeed good to see such practical applications shown in the literature.
> >
> > FWIW, I also wanted to mention that renaming your method to HNPE as you suggested in one of your responses seems like a good idea to me and should make the connection to SNPE more apparent.

---

### Author Response · Authors · 2021-08-09
**Main answer to all reviewers**

We would like to thank the reviewers for their insightful comments which will help us improve the article. Most reviewers judged our work ‘clearly written’, ‘relevant’ and a ‘valuable addition to the conference’, yet pointing some concerns related to our experimental validation.

Some reviewers thought that our experiments ‘could be made more convincing’ and we address this issue in our response to Reviewer 1 (tXAy). One of the main points that we wished to clarify is the fact that our work tackles a complete LFI pipeline, going from a theoretical neuroscience model (neural mass model) all the way to real physiological (EEG) recordings. Such full inferential analysis is very rare in the literature and is an important illustration of how one can actually use simulation-based inference to link real data with theoretical models. We believe that such experimental results are an important contribution to the community.

Other reviewers thought that the comparison of our $h$-Flow method versus a naive implementation was not fair due to differences in capacity and aggregation strategy; we address this issue in our response to Reviewer 2 (2NmZ) and Reviewer 3 (fnEF). In summary, we have rerun our experiments on the NMM with a fairer version of the naive architecture and have obtained the same qualitative results as in our initial submission.

Finally, we have also answered each one of the questions raised by the reviewers and clarified several points that will be improved in the camera ready version of our paper.

In light of our efforts for answering all questions raised by the reviewers and rerunning new experiments in line with their requests, we hope our answers will have resolved most of the issues raised in the reviews and that the reviewers will revise their initial rating in a fair and factual manner.

---

### Decision · Program_Chairs · 2021-09-27

**Decision:**

Accept (Poster)

**Comment:**

Dear authors,

congratulations on your paper being accepted at Neurips. As you know, the reviewers gave extensive comments and feedback on the submission, and we urge you to ensure that this feedback is incorporated into the final version of the article. In particular, reviewers felt that the name of the method, 'h-flow', is misleading, as the paper does not present a new flow, but rather a new SBI method (which will likely, but not necessarily, be applied to flows). We would ask you to take this advice seriously.

With best regards and congratulations, your AC